# Novel Strigolactone Mimics That Modulate Photosynthesis and Biomass Accumulation in *Chlorella sorokiniana*

**DOI:** 10.3390/molecules28207059

**Published:** 2023-10-12

**Authors:** Daria Gabriela Popa, Florentina Georgescu, Florea Dumitrascu, Sergiu Shova, Diana Constantinescu-Aruxandei, Constantin Draghici, Lucian Vladulescu, Florin Oancea

**Affiliations:** 1Bioproducts Team, Bioresources Department, National Institute for Research & Development in Chemistry and Petrochemistry—ICECHIM, Splaiul Independenței Nr. 202, Sector 6, 060021 Bucharest, Romania; daria.popa@icechim.ro (D.G.P.); diana.constantinescu@icechim.ro (D.C.-A.); 2Faculty of Biotechnologies, University of Agronomic Sciences and Veterinary Medicine of Bucharest, Bd. Mărăști Nr. 59, Sector 1, 011464 Bucharest, Romania; 3Enpro Soctech Com., Str. Elefterie Nr. 51, Sector 5, 050524 Bucharest, Romania; florentina.georgescu@enpro.ro (F.G.); lucian.vladulescu@enpro.ro (L.V.); 4“Costin D. Nenițescu” Institute of Organic and Supramolecular Chemistry, Romanian Academy, Splaiul Independentei Nr. 202B, Sector 6, 060023 Bucharest, Romania; cst_drag@yahoo.com; 5“Petru Poni” Institute of Macromolecular Chemistry, Romanian Academy, Aleea Grigore Ghica Voda Nr. 41-A, 700487 Iaşi, Romania; shova@icmpp.ro

**Keywords:** 3*H*-benzothiazol-2-one, 5*H*-furan-2-one, exogenous signals, quantum yield photosystem II, chlorophylls, proteins, microalgae biostimulants

## Abstract

In terrestrial plants, strigolactones act as multifunctional endo- and exo-signals. On microalgae, the strigolactones determine akin effects: induce symbiosis formation with fungi and bacteria and enhance photosynthesis efficiency and accumulation of biomass. This work aims to synthesize and identify strigolactone mimics that promote photosynthesis and biomass accumulation in microalgae with biotechnological potential. Novel strigolactone mimics easily accessible in significant amounts were prepared and fully characterized. The first two novel compounds contain 3,5-disubstituted aryloxy moieties connected to the bioactive furan-2-one ring. In the second group of compounds, a benzothiazole ring is connected directly through the cyclic nitrogen atom to the bioactive furan-2-one ring. The novel strigolactone mimics were tested on *Chlorella sorokiniana* NIVA-CHL 176. All tested strigolactones increased the accumulation of chlorophyll *b* in microalgae biomass. The SL-F3 mimic, 3-(4-methyl-5-oxo-2,5-dihydrofuran-2-yl)-3*H*-benzothiazol-2-one (**7**), proved the most efficient. This compound, applied at a concentration of 10^−7^ M, determined a significant biomass accumulation, higher by more than 15% compared to untreated control, and improved the quantum yield efficiency of photosystem II. SL-F2 mimic, 5-(3,5-dibromophenoxy)-3-methyl-5*H*-furan-2-one (**4**), applied at a concentration of 10^−9^ M, improved protein production and slightly stimulated biomass accumulation. Potential utilization of the new strigolactone mimics as microalgae biostimulants is discussed.

## 1. Introduction

Strigolactones (SLs) are carotenoid-derived compounds that act as signaling molecules with an increased importance in plant science. The first strigolactone was isolated from root exudates of cotton in 1966 by Cook, who named it strigol [1] and proposed its structure [2]. SLs have been identified as phytohormones with an important role in controlling the plant architecture [3,4,5,6,7,8,9,10,11,12,13] and in plant response to biotic and abiotic stress [14,15,16,17]. These molecules are exo-signals in the rhizosphere, triggering the germination of seeds of parasitic weeds [18,19,20,21,22,23,24,25], acting as branching factors for arbuscular mycorrhizal (AM) fungi [25,26,27], promotor of rhizobia nodulation [28,29,30], regulator of rhizosphere microbiome [31,32], plant neighbor detector [33,34,35]. All these properties stimulated the interest in identifying the various roles played by SLs in (parasitic plant) seed germination, plant development and its response to environmental factors, and molecular receptors involved in the signaling mechanism [36,37,38].

The initial discovered natural SLs (canonical SLs) contain a tricyclic lactone (ABC-rings) connected using an enol ether unit to a furan-2-one ring (D). Usually, these SLs (also called canonical) are classified based on the stereochemistry of the BC-ring junction into two major groups, one with the parent structure of strigol and the other with the parent structure of oronbanchol—Figure 1 [39].

The structure-activity relation studies showed that the presence of the bioactiphore CD part, the stereochemistry, and the methyl substituent at C-4′ of the D-ring are essential structural features for natural SLs activity as germination stimulants for the seeds of parasitic weeds [22,40]. The SL bioactivity is not influenced by the structural changes of A, B, and C rings or by the substitution of an oxygen atom by the nitrogen atom as in imino SL analogs [41] and strigolactam analogs [42], or by a sulfur atom [6].

In the last decade, new stimulants of the parasitic weed seed germination were discovered and included in the non-canonical strigolactone category [43]. Such strigolactones retain methylated butenolide D-ring but lacks the A-, B- or C-rings.

Only tiny amounts of natural SLs have been isolated from plant root exudates. These compounds have structures that are too complex for large-scale syntheses. Therefore, a significant number of simplified synthetic SL analogs and mimics were synthesized in order to understand their diverse structure-activities relationship and detect the corresponding receptors in plants. The SL analogs retain the bioactiphore part of natural SLs and have a considerable degree of molecular freedom in the AB-rings of the molecule [22,38,40,41,42,44,45,46,47,48,49,50,51,52]. In SL mimics, the bioactive D-ring is directly connected to aroyloxy, (hetero)aryloxy, arylthio, or (hetero)aromatic moieties. They mimic various biological activities of natural SLs even if they do not have the typical structural features of natural SLs [6,52,53,54,55,56,57,58]. SL analogs and mimics also have the potential for the development of practical application of strigolactones for parasitic weed control [22] or as plant biostimulants, to improve symbiosis efficiencies, to enhance crop tolerance to biotic and abiotic stress (including soil pollutants) and to regulate cultivated plant architecture [59,60].

It seems that the initial function of strigolactones in terrestrial ecosystems was as exo-signals for recruiting plant-beneficial microorganisms [61,62]. 5-Deoxystrigol determines an akin effect on aquatic ecosystems, inducing the formation of a symbiotic consortium between *Scenedesmus obliquus, Ganoderma lucidum,* and bacteria from activated sludge and enhancing their biotechnological performance for pollutant removal and biogas upgrading [63]. The strigolactone analog GR24 induced symbiotic pellet formation between *Chlorella vulgaris*, *G. lucidum,* and endophytic bacteria and improved their efficiency in nutrient removal from biogas slurry and fixation of CO_2_ from biogas stream [64,65,66]. GR24 has a similar boosting effect on photosynthetic activities in plants [67,68] and microalgae [64]. Therefore, SLs have potential applications in microalgae biotechnology as well. However, despite their application at very low concentrations, 5-deoxystrigol, and GR24 are too challenging to synthesize at the kilograms level required for large-scale application, and more affordable compounds are needed to exploit the SL potential for microalgae biotechnology.

Continuing our interest in novel bioactive compounds for photosynthetic organism development [58,69,70], we present here novel SL mimics, easily accessible in significant amounts from simple and commercially available starting compounds such as 3,5-disubstituted phenols or 2-hydroxybenzothiazole derivatives and 3-methyl-5*H*-furan-2-one. SL mimics derived from phenols, usually named debranones, have an extensive range of SL-like activity [6,53,54]. Benzothiazole is a privileged bicyclic ring system with numerous biological properties [71,72]. Novel SL-like bioactive compounds containing substituted aryloxy moieties connected to the bioactive 3-methyl-5*H*-furan-2-one ring, as well as a benzothiazole ring connected directly via the cyclic nitrogen atom to the bioactive 3-methyl-5*H*-furan-2-one ring have been synthesized and investigated for their effect on microalgae photosynthesis and biomass, photosynthetic pigments, and protein accumulations.

## 2. Results

### 2.1. Synthesis of New Strigolactone Mimics

The first two SL mimics are derived from 3,5-disubstituited phenols and contain the bioactive 3-methyl-5*H*-furan-2-one ring linked to an aryloxy moiety. They have been synthesized with good yields by the coupling reactions of 3,5-disubstituted phenols with a common intermediate 5-bromo-3-methyl-5*H*-furan-2-one prepared by the bromination of 3-methyl-2-furanone [73]. Thus, the reaction of 3,5-dimethylphenol **1a** with the 5-bromo-3-methyl-5*H*-furan-2-one **2** in acetone, in the presence of potassium carbonate, led to SL mimic SL-F1, 5-(3,5-dimethylphenoxy)-3-methyl-5*H*-furan-2-one **3**. Under the same reaction conditions, 3,5-dibromophenol **1b** reacts with 5-bromo-3-methyl-5*H*-furan-2-one **2** to give the SL mimic SL-F2, 5-(3,5-dibromophenoxy)-3-methyl-5*H*-furan-2-one **4** (Figure 1). The structures of these new SL mimics **3** (SL-F1) and **4** (SL-F2) (See Appendix B, Table A1) have been confirmed by chemical and spectral analyses (FTIR spectroscopy ^1^H NMR, ^13^C NMR) shown in the Appendix A.

The next SL mimics contain a bioactive benzothiazole unit connected to the bioactive 3-methyl-5*H*-furan-2-one structure. The starting compound, commercially available 2-hydroxybenzothiazole **5**, could have enol–keto tautomeric forms, usually the keto tautomeric form being the most stable. Consequently, two different compounds could be formed in the reaction of 2-hydroxybenzothiazole **5** with 5-bromo-3-metyl-furan-2-one **2**. The first possible SL mimic, 5-(benzothiazol-2-yloxy)-3-methyl-5*H*-furan-2-one **6**, in which the two bioactive structural units are connected via an ether link, is the result of the coupling reaction of the enol-tautomeric form **5a** with the intermediate **2**. The second one, 3-(4-methyl-5-oxo-2,5-dihydrofuran-2-yl)-3*H*-benzothiazol-2-one **7**, in which the 3-methyl-5*H*-furan-2-one unit is directly connected with the nitrogen atom from the benzotiazole ring, is the alkylation product of keto-tautomeric form **5b** with the intermediate **2.** By the reaction of commercial product 2-hydroxybenzothiazole **5** with 5-bromo-3-methyl-5*H*-furan-2-one **2** in anhydrous acetone in the presence of anhydrous potassium carbonate at room temperature, we have isolated and completely characterized only the *N*-alkylation product SL-F3, 3-(4-methyl-5-oxo-2,5-dihydrofuran-2-yl)-3*H*-benzothiazol-2-one **7** derived from the keto-form, 3*H*-benzothiazol-2-one **5b** (Figure 2).

The next starting compound, 6-bromo-3*H*-benzothiazol-2-one **8**, was obtained by bromination with molecular bromine of the commercial product 3*H*-benzothiazol-2-one **5b**. The bromination was performed with molecular bromine in glacial acetic acid at room temperature, an improved procedure for the synthesis of bromine-substituted benzothiazoles [74]. The regioselectivity of bromination was deduced by NMR spectroscopy, and the structure was confirmed by single-crystal X-ray analysis. According to X-ray crystallography (Appendix A), compound **8** crystallizes in the *C*2/*c* space group of a monoclinic system with one molecule as an asymmetric unit. It is worth mentioning that each pair of centrosymmetrically related molecules is associated with a dimeric derivative via the cyclic H-bonding synthon, where N-H acts as a donor while the oxygen atom is the acceptor of the proton. The further interaction of the dimeric units occurs via π-π stacking with the centroid-to-centroid distance of 3.9602(3) Å. It determines the formation of the one-dimensional supramolecular array, as shown in Figure 2b.

The reaction of the common intermediate 5-bromo-3-methyl-5*H*-furan-2-one **2** with 6-bromo-3*H*-benzothiazol-2-one **8** led easily to the *N*-alkylation product SL-F5, 6-bromo-3-(4-methyl-5-oxo-2,5-dihydrofuran-2-yl)-3*H*-benzothiazol-2-one **9** (Figure 3). In the same way, the reaction of the 5-bromo-3-methyl-5*H*-furan-2-one **2** with 6-nitro-3*H*-benzothiazol-2-one **10** [75] prepared by the reaction of 3*H*-benzothiazol-2-one **5b** with nitric acid in acetic acid, gave the *N*-alkylation product SL-F6, 6-nitro-3-(4-methyl-5-oxo-2,5-dihydrofuran-2-yl)-3*H*-benzothiazol-2-one **11** (Figure 3).

The structures of *N*-alkylation products **7** (SL-F3), **9** (SL-F5), and **11** (SL-F6) (see Appendix B, Table A1) have been confirmed by chemical and spectral analysis. The IR spectra (Appendix A) showed the presence of two carbonyl bands at about 1755–1769 cm^−1^ (C=O from furanone ring) and 1673–1690 cm^−1^ (C=O from benzothiazole ring). The ^1^H NMR spectra of these compounds revealed the presence of all characteristic protons (Appendix A). ^13^C NMR spectra clearly indicated the presence of two carbonyl carbons at about 169.1–170.0 ppm and 170.5–171.2 ppm for all these SL mimics (Appendix A). The alkylation with bromolactone 2 at the nitrogen atom of the benzothiazole ring was confirmed by the single-crystal X-ray analysis for compound **7** (SL-F3). The results of the X-ray diffraction study for seven are presented in Figure 3a and Appendix A.

The crystal structure is based on an extended system of C-H···O hydrogen bonding. As shown in Figure 3b, each molecule is interacting via six H-bonds with six adjacent symmetrically related molecules, which determined the formation of two-dimensional supramolecular layers parallel to the 011 plane. A view of the 2D supramolecular architecture is depicted in Figure 3c. The crystal structure of compound **7** (SL-F3) is built up from the packing of discrete supramolecular layers (Appendix A).

### 2.2. The Biological Activity of Synthesized Compounds

The microalgae growth was monitored by reading the optical density at 750 nm. Figure 4 illustrates the dynamics of the optical densities at four representative moments of tested microalgae growth at 2, 5, 9, and 14 days of cultivation.

Significant differences between treatments were observed since the fifth day of *Ch. sorokiniana* NIVA-CHL 176 microalgae cultivation in the presence of the newly synthesized SL mimics at two concentrations, 10^−7^ and 10^−9^ M, respectively. These concentrations were selected based on preliminary trials that corroborated the existing data on the effects of natural strigolactone, 5-deoxystrigol, and SL analog GR 24 on microalgae [63,64]. The new mimic SL-F5, applied at a concentration of 10^−9^ M (V4c2), recorded the significantly highest absorbance compared to the solvent control (SC). The mimic SL-F3 applied at 10^−7^ M (V3c1) also determined a significant stimulation of microalgae growth compared to SC. The rest of the treatments, including GR24, had a marginally significant growth stimulation.

Between 5 and 9 days of culturing, the SL mimic treatments were less effective in stimulation of microalgae. At the 9th day of growth, most treatments did not record statistically significant differences between treatments and SC, except the mimic SL-F2 applied at 10^−7^ M (V2c1), which determined a marginally significant stimulation compared to the effect of solvent applied at 10^−7^ M (SC).

At the end of the culturing period, at 14 days, the cell growth upon some treatments was significantly higher, while upon others, the cell growth was slightly lower than in controls. The treatment that stood out as inducing the highest significant cell growth was V3c1, i.e., the strigolactone mimic SL-F3, at the higher concentration tested, 10^−7^ M, stimulated microalgae cell growth the most.

The SL analog GR24 did not determine statistically relevant effects on *Ch. sorokiniana* NIVA-CHL 176 after 9 and 14 days. The mimic SL-F6 has a dual response. Applied at the higher concentration, 10^−7^ M (V5c1), this mimic showed a marginally significant stimulative effect after 14 days. Applied at the lower concentration, 10^−9^ M, SL-F6 determined an inhibition of microalgae development that is statistically relevant compared to itself at the higher concentration and to SC.

We measured the chlorophyll fluorescence as an indicator of photosynthetic energy conversion in microalgae (Figure 5) at the end of the cultivation period (after 14 days). The results pointed towards SL-F3 as being the most effective in microalgae stimulation. At the higher concentration tested, 10^−7^ M, this SL mimic showed a statistically significant improvement in the functioning of system II in tested microalgae cells. The positive effect was at the limit of statistical significance for SL-F3 at 10^−9^ M and for SL-F5 at 10^−9^ M.

The content of chlorophyll *a* (ChlA), chlorophyll *b* (ChlB), total carotenoids, and total photosynthetic pigments are presented in Figure 6.

Regarding chlorophyll *a* (Figure 6a), a significant stimulatory effect was observed when treating *Ch. sorokiniana* with SL-F5 at the lower concentration, 10^−9^ M, (V4c2), compared to the controls. At 10^−7^ M concentration, two other strigolactone mimics, SL-F3 (V3c1) and SL-F2 (V2c1), induced the next higher ChlA than the rest of the variants and determined a slight increase in ChlA, but not statistically significant compared to SC.

Chlorophyll *b*, usually found in small amounts in microalgae cells, varied from 0.1 mg/L in controls to 1.4 mg/L in V4c2 (SL-F5). These variations (Figure 6b) show that all tested synthetic strigolactones, GR24 analogs, and the newly synthesized SL mimics determined a statistically significant increase in chlorophyll *b* accumulation in the biomass of treated microalgae.

On the other hand, the impact of SL mimics on the total carotenoids extracted from *Ch. sorokiniana* (Figure 6c) was not significant. However, in the case of SL-F3 applied at the higher concentration, 10^−7^ M (V3c1), the highest carotenoid content among all variants was obtained.

In the case of the total pigment accumulation (Figure 6d), the highest quantity of pigments, more than 6.5 mg/L, was found in the culture treated with SL-F5 at 10^−9^ M(V4c2)—61.8% increase compared to C and CS (approx. 4 mg/L). ChlA and ChlB have the largest share in this sum, significantly increasing compared to controls. A similar higher total pigment accumulation was observed for the treatments with SL-F1, SL-F2, and SL-F3 at the higher concentration tested (10^−7^ M)—with an increase of 38.5%, 48.3%, and 43.8%, respectively, compared to controls. These newly synthesized SL mimics showed higher stimulatory effects than the strigolactone analog (GR24, c1, and c2).

The average quantities of microalgae biomass obtained for each treatment are represented in Figure 7, expressed as grams of dry biomass per liter of growing media.

The results were similar to those determined when the optical densities were used to estimate the cell dynamics and accumulation of *Ch. sorokiniana*. In the case of microalgae in the genus *Chlorella*, light attenuation is determined mainly by scattering, which is at least 50 times higher than the absorption from photosynthetic pigments [76]. The determination of dried biomass confirmed the stimulatory effect of SL-F3 at the concentration of 10^−7^ M (V3c1) in *Ch. sorokiniana*. This increase was more than 20% compared to other SL mimics that determined a slight inhibition of biomass accumulation, i.e., SL-F1 applied at a lower concentration, 10^−9^ M (V1c2) and SL-F2 applied at both concentrations (V2c1 and V2c2). The higher biomass accumulation was also significant compared to controls (around 11%) and GR24 reference strigolactone analog. The next highest biomass production was induced by V5c1, similar to optical density, but the difference in biomass production was not statistically significant compared to the controls.

We determined the soluble proteins from the dried microalgae biomass using Bradford and Biuret, shown in Figure 8 and Figure 9, respectively.

There was a significant difference between soluble proteins determined by Biuret and Bradford methods. This difference is due to the limitations of these methods used for protein quantification in microalgae [77]. The Bradford assay does not detect peptides with a molecular mass lower than 3–5 KDa [78] and underestimates the soluble protein content in microalgae [79]. The Biuret method interferes with polyphenols [80] and starch [81] and overestimates the soluble protein content in microalgae [82]. Despite these limitations, both methods indicate that SL-F2 (V2c2), applied at the lower concentration (10^−7^ M), significantly increases soluble protein accumulation compared to both C and SC. Other differences in protein concentrations were obtained for the treatments with SL mimics derived from benzothiazoles, SL-F3 and SL-F6, especially when applied at the lower concentration, 10^−9^ M, and SL-F1 at 10^−7^ M, that stimulated accumulation of Bradford-reactive soluble proteins, without influencing or slightly decreasing the content of compounds giving biuret reaction. These differences are influenced by the effect of the solvent used for SL solubilization (DMSO—SC), which significantly enhanced the accumulation of the Bradford reactive proteins compared to the non-treated control (C)—by 12.8%.

## 3. Discussion

Strigolactones are complex molecules with multiple biological functions and high molecular diversity [83,84]. The natural strigolactones are challenging to extract and purify from plant tissues [85]. Their preparation from root exudates of plants cultivated in aeroponic conditions is limited to the species-specific exuded strigolactones—e.g., maize-produced non-canonical zealactones when cultivated in aeroponic conditions [86].

The chemical synthesis of canonical natural strigolactones is challenging [87], and preparing the ABC scaffolds requires a long route, which makes it hard to upscale to multigram synthesis. The chemical synthesis of non-canonical SLs is also challenging, involving numerous steps, with a low SL final yield, and starting from highly purified organic compounds that are laborious to prepare [88]. For example, a synthesis of carlactone, a non-canonical strigolactone that is also an intermediary in the biosynthesis of canonic strigolactones, starting from 2,6-dimethylcyclohexanone, requires seven steps, with a final yield of 0.4% [89].

As was mentioned, synthetic SL analogs and mimics were previously proposed, being easier to prepare than natural SLs and with significant biological activity [90]. For the most used strigolactone analog, GR24, a synthetic route suitable for multigram preparation, was developed [91]. However, the cost for the synthesis of GR24 is still high, and new synthetic SL analogs and mimics are needed for practical applications of strigolactones [59,87].

The target of this work was to prepare novel strigolactone mimics easily accessible in significant amounts from simple and available starting materials that could be used as microalgae biostimulants. The first two novel compounds contain 3,5-disubstituted aryloxy moieties connected to the bioactive furan-2-one ring. A benzothiazole ring connected directly via the cyclic nitrogen atom to the bioactive furan-2-one ring is characteristic of the second group of compounds. The newly synthesized SL mimics modulate the photosynthesis and biomass accumulation in the *Ch. sorokiniana* NIVA-CHL 176 strain. *Ch. sorokiniana* NIVA-CHL 176 is a robust microalgal strain that has the potential for phytoremediation [92], carbon sequestration [93], and functional food/feed production [94].

All tested strigolactones, GR24 analog, and the newly synthesized SL mimics significantly increase the synthesis of chlorophyll *b*. Chlorophyll *b* enhances the photosynthesis efficiency in microalgae [95], and in *Ch. vulgaris*, it was shown to act as a photoprotecting pigment [96]. The SL-F3 mimic, 3-(4-methyl-5-oxo-2,5-dihydrofuran-2-yl)-3*H*-benzothiazol-2-one (compound **7**), proved to be the most efficient microalgae biostimulant. This compound, applied at a concentration of 10^−7^ M, determined a significant biomass accumulation, higher by more than 15% compared to untreated control, and also improved the quantum yield efficiency of photosystem II. SL-F2 mimic, 5-(3,5-dibromophenoxy)-3-methyl-5*H*-furan-2-one (compound **4**), applied at a concentration of 10^−9^ M, improved protein production and slightly stimulated biomass accumulation.

“Microalgae biostimulants” is a category of products that has similar functions on microalgae biotechnology that plant biostimulants have in agriculture. Plant biostimulants are agricultural inputs that fulfill one of the following agricultural functions: enhance/benefit nutrient uptake and utilization, increase cultivated plant tolerance to abiotic stress, and improve crop quality traits due to an increased accumulation of (phyto)nutrients in harvest [97]. The mirror concept of “microalgae biostimulants” is still new and not yet largely used. Our group described as ”microalgae biostimulants” the humic acids [98,99] and selenium—betaine combination [100] that determine microalgae effects analog to those exerted by plant biostimulants on crops. An international group centered at Malaysian universities utilizes the term ”microalgae biostimulants” to describe the effects of feather hydrolysate [101] and antioxidant extract from onion peels [102] on microalgae. The functions defined for microalgae biostimulants could significantly contribute to the development of microalgae technology.

Abiotic stress, including light stress, is used to trigger the accumulation of compounds of interest in microalgae [103,104]. However, such stressful condition limits microalgae productivity and biosynthesis efficiency [105]. Phytohormones were proposed to optimize the application of the stress conditions while maintaining microalgae photosynthetic productivity [106,107]. Microalgae biostimulants seem to be more efficient tools than phytohormones to balance the increased accumulation of the compounds of interest in stress conditions and photosynthetic efficiency. By definition, their function is to increase microalgae tolerance to abiotic stress and to promote the accumulation of compounds of interest (“microalgae biomass quality”).

Strigolactones are not known to be present in microalgae, neither as phytohormones nor as quorum-sensing signals. Strigolactones first appeared in terrestrial plants, in bryophytes, acting as quorum-sensing signals [108] and for the recruitments of the mycorrhizae fungi [62,109]. The non-canonical strigolactones are present in larger quantities in root exudates and are mainly involved in rhizosphere signaling [43,110]. The canonical strigolactone evolved as hormonal signaling in flowering plants, maintaining their function as rhizosphere exo-signals [61]. The presence of the ABC rings and the stereochemistry seem to be essential for the proliferation of AM fungi hyphae [8,27]. In rice, canonical strigolactones are more efficient as rhizosphere signals than determinants of tillering [111]. The root-exuded strigolactones act similarly to a quorum-sensing signal, being a cue for the presence of the neighboring plants [33,34]

The function of strigolactones in terrestrial plants seems to be as an integrator of metabolic and nutritional signals [83] and an orchestrator of response to abiotic stress [16,17]. For example, strigolactones modulate the synthesis of chlorophyll *b* in a stress-specific manner. In maize plants subject to drought stress (and consequent oxidative stress), strigolactones increase the synthesis of chlorophyll *b*, which protects the photosystems against (photo)oxidative stress [112]. In cucumber plant subjected to low light stress (and without photooxidative stress), the strigolactone application decreases the chlorophyll *b* synthesis, promoting the synthesis of the more efficient chlorophyll *a* pigment [113]. A similar modulating effect was shown in the present work for microalgae submitted to a continuous light stress (100 μmol/m^2^·s)—an increase in the synthesis of the chlorophyll *b* photoprotective pigment in the case of continuous illumination and photooxidative stress. The continuous illumination for a photoperiod longer than 2 min was proved to determine light stress in *Ch. vulgaris* strain SAG 12A, reducing the efficiency of light harvesting systems [96].

The application of exogenous GR24 to *Artemisia annua* increased the quantum yield efficiency of the photosystem II, determined by chlorophyll fluorescence, Fv/Fm [114]. Our results showed a similar effect for SL mimics derived from benzothiazoles. The SL-F3 mimic, 3-(4-methyl-5-oxo-2,5-dihydrofuran-2-yl)-3*H*-benzothiazol-2-one (compound **7**), has the highest effect, statistically significant compared to control and other SL treatments, in enhancing the quantum yield efficiency of photosystem II. Chlorophyll fluorescence is a an efficient tool for non-invasive monitoring microalgae photosynthetic efficiency in various conditions [115]. We used the pulse-amplitude modulated (PAM) chlorophyll fluorescence induction, i.e., assay of the chlorophyll fluorescence after adaptation to dark (no active photosynthetic centers) and after a saturation pulse—all photosynthetic centers saturated after an actinic pulse [116]. The quantum yield efficiency of photosystem II provides information related to the integrity and photochemical efficiency of the photosynthetic apparatus [117].

In plants, the application of the strigolactone analog GR24 determines an improvement in crop quality traits. In corn, the foliar application of GR24 in saline condition stress increased the number of grain per cob and the cob diameter [118]. In *Artemisia annua* the application of GR24 leads to an increased artemisin accumulation [114]. In our experiments, SL-F2 applied at the lower concentration (10^−7^ M), significantly increased the soluble protein accumulation (determined by the Bradford and Biuret methods). The treatments with SL mimics derived from benzothiazoles, SL-F3 and SL-F6, especially at the lower concentration, 10^−9^ M, and SL-F1 at 10^−7^ M, stimulate accumulation of the Bradford-reactive soluble proteins. These characteristics are important considering the potential use of *Ch. sorokiniana* NIVA-CHL 176 for production of functional foods/feeds.

Previous studies demonstrated the effects of the strigolactones analog GR24 [64,66,119] and of a natural strigolactone, 5-deoxystrigol [63], i.e., molecules that include A, B, C, and D rings, on microalgae,. The present study demonstrates, for the first time to the best of our knowledge, that strigolactone mimics, retaining only the butenolide D ring, are also active on microalgae. This study has practical applicationa, the newly synthesized SL mimics being candidates for the development of efficient microalgae biostimulant formulations. However, the mechanisms behind these results, through which the novel SLs mimics biostimulate microalgae needs further investigations. There are no studies related to strigolactone receptors from microalgae or related to the biological significance of the microalgae response to signaling molecules specific to terrestrial plants.

## 4. Materials and Methods

### 4.1. Materials

To assesses the biological effects of the newly synthesized strigolactones we used the *Chlorella sorokiniana* NIVA-CHL 176 strain, from the Norwegian Culture Collection of Algae, NORCCA. The reference strigolactone analog racemic (±) GR24 was purchased from StrigoLab (Turin, Italy). For preparing the BG11 growth media, we used chemicals with quality appropiate for the cultivation of the microalgae. Sodium nitrate (NaNO_3_), magnesium sulphate (MgSO_4_·7H_2_O), Calcium chloride (CaCl_2_·2H_2_O), Iron(III) chloride, cobalt(II) nitrate (Co(NO_3_)_2_·6H_2_O) were purchased from Merck (Merck Group, Darmstadt, Germany), potassium dihydrogen phosphate (KHPO_4_), citric acid, boric acid (H_3_BO_3_), zinc sulphate (ZnSO_4_·7H_2_O), sodium molybdate (Na_2_MoO_4_·2H_2_O) and cobalt(II) nitrate (Co(NO_3_)_2_·6H_2_O were supplied by Scharlau (Scharlab, Barcelona, Spain), EDTA disodium salt Na_2_EDTA·2H_2_O was purchased from Fluka (Honeywell, Morris Plains, NJ, USA), Manganese chloride (MnCl_2_·4H_2_O) was supplied by Carl Roth (Karlsruhe, Germany), copper sulphate (CuSO_4_·5H_2_O) was purchased from Chimopar. (Bucharest, Romania). The reagents used for biochemical analysis and extractions were analytical quality (p.a.). Dimethyl sulfoxide (DMSO) was purchased from Merck (Darmstadt), sodium hydroxide was supplied by Chimopar (Bucharest), and Bovine serum albumin (BSA) was purchased from Carl Roth (Karlsruhe).

### 4.2. General Information

The common intermediate, 5-bromo-3-methyl-5*H*-furan-2-one **2**, was obtained with good yield by the bromination of 3-methyl-5*H*-furan-2-one with *N*-bromosuccinimide in CCl_4_, in the presence of benzoyl peroxide [73] and it was used as crude reaction product (94–95%).

The melting points (mp) were determined on a Boetius apparatus and are uncorrected. The IR spectra were registered on a Fourier-transform (FT)-IR Vertex 70 spectrometer (Bruker Optik GmbH, Ettlingen, Germany) in ATR modes. The NMR spectra were recorded on a Varian Gemini 300 BB instrument, operating at 300.1 MHz and 75.5 MHz for ^1^H and ^13^C nuclei, respectively, or on a Bruker Avance III instrument, operating at 500 MHz, and 100 MHz, respectively. Chemical shifts are reported as δ (ppm) and were referenced to internal TMS for ^1^H chemical shifts and to the internal deuterated solvent for ^13^C chemical shifts (CDCl3 referenced at 77.0 ppm) and unambiguously assigned based on additional COSY, HSQC/HETCOR and HMBC experiments. The elemental analysis was carried out on a COSTECH Instruments EAS32 apparatus. Satisfactory microanalyses for all new compounds were obtained.

### 4.3. General Procedure for Preparation of SL Mimics Derived from 3,5-Disubstituted Phenols

To a solution of 3,5-disubstituted phenol **1a** or **1b** (10 mmol) in 20 mL acetone anhydrous potassium carbonate (1.52 g, 11 mmol) was added, then crude 5-bromo-3-methyl-5*H*-furan-2-one 2 (2.21 g, 12.7 mmol) was slowly added under stirring at room temperature. The stirring was continued for 16 h at room temperature. After cooling, the reaction mixture was filtered off to remove the potassium carbonate and the solvent was evaporated. The residue was chromatographed on a silicagel/alumina 70–230 mesh packed column eluting with dichloromethane and the final compounds were obtained after the complete removal of the eluting solvent.

*5-(3,5-Dimethylphenoxy)-3-methyl-5H-furan-2-one* (**3**) [SL-F1]. Viscous, pale yellow liquid after purification using column chromatography. Yield 64%. Anal. Calcd. for C_13_H_14_O_3_ (218.25): C 71.54, H 6.47%; Found: C 71.62, H 6.56%. ATR-FTIR (solid): 1771 cm^−1^ (νC=O). ^1^H NMR (300 MHz, CDCl_3_) δ (ppm): 2.12 (s, 3H, Me), 2.44 (s, 6H, 2Me, phenyl), 6.38 (s, 1H, H-5), 6.87 (3H, H-4, 2H phenyl), 7.10 (s, 1H, phenyl). ^13^C NMR (75 MHz, CDCl_3_) δ (ppm): 10.5 (Me), 21.3 (2Me, phenyl), 99.1 (C-5), 114.4 (C-2, C-6, phenyl), 125.1(C-4, phenyl), 133.9 (C-3), 139.4 (C-3, C-5, phenyl), 142.6 (C-4), 156.4 (C-1, phenyl), 171.4 (C=O).*5-(3,5-Dibromophenoxy)-3-methyl-5H-furan-2-one* (**4**) [SL-F2]. The compound was crystallized from cyclohexane as white crystals with mp 88–91 °C; yield 62%. Anal. Calcd. for C_11_H_8_Br_2_O_3_ (347.99): C 37.97, H 2.32%; Found: C 38.91, H 2.40%. ATR-FTIR (solid): 1776 cm^−1^ (νC=O). ^1^H NMR (300 MHz, CDCl_3_) δ (ppm): 2.02 (t, 3H, Me, *J* = 1.4 Hz), 6.23 (quintet, *J* = 1.4 Hz, 1H, H-5), 6.97 (quintet, *J* = 1.5 Hz, 1H, H-4), 7.24 (d, 2H, *J* = 1.4 Hz, H-2, H-6, phenyl), 7.24 (t, 1H, *J* = 1.4 Hz, H-4 phenyl). ^13^C NMR (75 MHz, CDCl_3_) δ (ppm): 10.8 (Me), 98.5 (C-5), 113.2 (C-2, C-6, phenyl), 123.3 (C-3, C-5, phenyl), 129.4 (C-4, phenyl), 134.9 (C-3), 141.7 (C-4), 157.3 (C-1, phenyl), 170.9 (C=O).

### 4.4. General Procedure for Preparation of SL Mimics Derived from Benzothiazoles

6-Bromo-3*H*-benzothiazol-2-one **8** was prepared by treating the commercially product 3*H*-benzothiazol-2-one **5** with bromine in acetic acid [74]. 5-Nitro-3*H*-benzothiazol-2-one **10 [75]** was prepared by treating the commercially product 3*H*-benzothiazol-2-one **5** with nitric acid in acetic acid.

Anhydrous potassium carbonate (1.52 g, 11 mmol) was added into a solution of (10 mmol) 2-hydroxybenzothiazole **5**, 6-bromo-2-hydroxybenzothiazole **8** or 6-nitro-2-hydroxybenzotiazole **10** in 20 mL acetone, then crude 5-bromo-3-methyl-5*H*-furan-2-one **2** (2.21 g, 12.7 mmol) was slowly added under stirring at room temperature. The stirring was continued for 16 h at room temperature. After cooling, the reaction mixture was filtered off to remove the potassium carbonate and the solvent was evaporated. The residue was chromatographed on an alumina-packed column eluting with dichloromethane and the final compounds were crystallized from a suitable solvent.

*3-(4-Methyl-5-oxo-2,5-dihydrofuran-2-yl)-3H-benzothiazol-2-one* (**7**) [SL-F3]. The compound was crystallized from 2-propanol as white crystals with mp 153–155 °C; yield 42%. Anal. Calcd. for C_12_H_9_NO_3_S (M 247.27): C 58.29, H 3.67, N 5.66%; Found: C 58.21, H 3.42, N 5.71%. ATR-FTIR (solid): 1764 cm^−1^ (νC=O, lactone), 1673 cm^−1^ (νC=O, lactam). ^1^H NMR (300 MHz, CDCl_3_) δ (ppm): 2.28 (s, 3H, Me), 7.07 (d, 1H, *J* = 7.7 Hz, benzothiazole), 7.16 (bs, 1H, H-5), 7.35–7.42 (m, 3H, H-4 and 2H-benzothiazole), 7.58–7.60 (m, 1H, benzothiazole). ^13^C NMR (75 MHz, CDCl_3_) δ (ppm): 10.5 (Me), 81.5 (C-5), 111.9 (C-4, benzothiazole), 122.0 (C-7a, benzothiazole), 123.0, 124.2, 126.8 (C-5, C-6, C-7, benzothiazole), 134.4 (C-3), 135.0 (C-3a, benzothiazole), 143.1 (C-4), 170.0, 171.2 (2 C=O).*6-Bromo-3-(4-methyl-5-oxo-2,5-dihydrofuran-2-yl)-3H-benzothiazol-2-one* (**9**) [SL-F5]. The compound was crystallized from 2-propanol as white crystals with mp 176–177 °C, yielding 44%. Anal. Calcd. for C_12_H_8_BrNO_3_S (M 326.17): C 44.19, H 2.47, N 4.29%; Found: C 44.14, H 2.53, N 4.34. ATR-FTIR (solid): 1755 cm^−1^ (νC=O, lactone), 1690 cm^−1^ (νC=O, lactam). ^1^H NMR (300 MHz, CDCl_3_) δ (ppm): 2.27 (t, 3H, Me, *J* = 1.9 Hz), 6.94 (d, 1H, H-4, *J* = 8.8 Hz, benzothiazole), 7.10 (quintet, *J* = 1.9 Hz, 1H, H-5), 7.34 (quintet, *J* = 1.6 Hz, 1H, H-4), 7.53 (dd, 1H, H-5, *J* = 8.8, 1.9 Hz, benzothiazole), 7.72 (d, 1H, H-7, *J* = 1.9 Hz, benzothiazole). 13C NMR (75 MHz, CDCl_3_) δ (ppm): 11.1 (Me), 81.5 (C-5), 113.1 (C-4, benzothiazole), 117.0 (C-6, benzothiazole), 124.0 (C-7a, benzothiazole), 124.5, 128.8 (C-5, C-7, benzothiazole), 134.0 (C-3), 134.6 (C-3a, benzothiazole), 142.7 (C-4), 169.2, 170.9 (2 C=O).*6-Nitro-3-(4-methyl-5-oxo-2,5-dihydrofuran-2-yl)-3H-benzothiazol-2-one* (**11**) [SL-F6]. The compound was crystallized from acetonitrile as white crystals with mp 183–185 °C, yielding 39%. Anal. Calcd. for C_12_H_8_N_2_O_5_S (M 292.27): C 49.31, H 2.76, N 9.58%; Found: C 49.52, H 2.87, N 9.34%. ATR-FTIR (solid): 1769 cm^−1^ (νC=O, lactone), 1730 cm^−1^ (νC=O, lactam). ^1^H NMR (500 MHz, CDCl_3_) δ (ppm): 2.13 (t, 3H, Me, *J* = 1.7 Hz), 6.98 (t, *J* = 1.8 Hz, 1H, H-5), 7.05 (d, 1H, H-4, *J* = 9.0 Hz, benzothiazole), 7.23 (t, *J* = 1.7 Hz, 1H, H-4), 8.19 (dd, 1H, H-5, *J* = 9.0, 1.3 Hz, benzothiazole), 8.36 (d, 1H, H-7, *J* = 1.3 Hz, benzothiazole). ^13^C NMR (100 MHz, CDCl_3_) δ (ppm): 10.9 (Me), 81.4 (C-5), 111.6 (C-4, benzothiazole), 118.0 (C-5, benzothiazole), 122.8 (C-7, benzothiazole), 123.3 (C-7a, benzothiazole), 134.9 (C-3), 139.6 (C-3a, benzothiazole), 142.1 (C-4), 144.1(C-5, benzothiazole), 169.1, 170.5 (2 C=O).

### 4.5. Single Crystal X-ray Diffraction

The X-ray diffraction measurements were carried out with a Rigaku Oxford—Diffraction XCALIBUR E CCD diffractometer (Rigaku, Tokyo, Japan) equipped with graphite-monochromated MoKα radiation. The unit cell determination and data integration were carried out using the CrysAlis package of Oxford Diffraction (CrysAlisPro Software system, version 1.171.41.64, Rigaku Corporation, Oxford, UK). The structures were solved by Intrinsic Phasing using Olex2 software version 1.5 [120] with the SHELXT structure solution program [121] and refined using full-matrix least-squares on F2 with SHELXL-2015 [122] using an anisotropic model for non-hydrogen atoms. All H atoms were introduced in idealized positions (dCH = 0.96 Å) using the riding model. The H-atom attached to nitrogen were localized from difference Fourier maps, and their positional parameters were verified according to H-bonds geometry. The molecular plots were obtained using the Olex2 program. The crystallographic data and refinement details are quoted in Appendix A, while bond lengths and angles are summarized in Appendix A.

### 4.6. Biological Activities Tests

#### 4.6.1. Bioassay of Strigolactones Mimics

The *Ch. sorokiniana* NIVA-CHL 176 strain was cultivated in the presence of the strigolactone mimics, and several parameters specific for microalgae growth and development, and biotechnological application were determined: absorbance at 750 nm, dry biomass, photosynthetic pigments, soluble protein content.

#### 4.6.2. Growth Media

Microalgae cultures were grown in BG11 nutrient medium [123], which was prepared using stock solutions, the final concentration being: 17.6 mM NaNO_3_, 0.23 mM KHPO_4_, 0.3 mM MgSO_4_·7H_2_O, 0.24 mM CaCl_2_·2H_2_O, 0.31 mM Citric Acid·H_2_O, 0.021 mM FeCl_3_, 0.19 mM Na_2_EDTA·2H_2_O, 0.046 mM H_3_BO_3_, 9mM MnCl_2_·4H_2_O, 0.77mM ZnSO_4_·7H_2_O, 1.6 mM Na_2_MoO_4_·2H_2_O, 0.3 mM CuSO_4_·5H_2_O, 0.17 mM Co(NO_3_)_2_·6H_2_O. The medium was sterilized in an autoclave (MLS 375 1L, PHCBi, Wood Dale, IL, USA) at 121 °C for 15 min, and the Erlenmeyer flasks used for culturing were sterilized in a universal laboratory oven (UN 75, Memmert, Schwabach, Germany) at 180 °C for 3 h.

#### 4.6.3. Treatment Solutions

To solubilize the analog strigolactones SL-F1 (V1), SL-F2 (V2), SL-F3 (V3), SL-F5 (V4), and SL-F6 (V5), we used dimethyl sulfoxide (DMSO) as solvent. Five stock solutions of 1 mM were prepared, considering the molar mass of each chemical. The treatments were done at two SL concentrations: the final concentrations tested were 1 × 10^−7^ M (c1) and 1 × 10^−9^ M (c2), prepared from the stock solution by dilutions with BG11 medium. We also included a solvent control (SC, 10^−7^ M DMSO) and a reference product, strigolactone analog GR24 (C-GR24). The treatments were assessed in triplicate in a final culturing volume of 25 mL in 50 mL sterilized Erlenmeyer flasks.

#### 4.6.4. Culturing Conditions

The growth medium was inoculated at a final concentration of 1% microalgae biomass with the microalgae *Ch*. *sorokiniana* using a fresh culture in a laminar flow microbiological hood (Bio 2 Advantage Plus, Azbil Telstar, Terrassa, Spain) to avoid contamination. The cultures were kept in a growth chamber (AlgaeTron AG-230-ECO, Photon Systems Instruments, Drásov, Czech Republic) under controlled conditions of temperature and light (25° ± 1° C, illumination with white, fluorescent lamp at 120 μmol/m^2^·s, with a light/dark photoperiod of 14/10 h) and under continuous orbital shaking of 130–150 rpm (Unimax 1010 orbital shaker, Heidolph, Schwabach, Germany), for 14 days.

#### 4.6.5. Growth Parameters

We determined the microalgae growth by measuring the optical density on a microplate reader (CLARIOStar, BMG Labtech, Ortenberg, Germany) at 750 nm of the aseptically taken samples from microalgae culture [124]. The readings were performed in sterile 96-well plates in triplicates.

#### 4.6.6. Biomass

A volume of 22 mL of *Ch*. *sorokiniana* cultures was added to a pre-weighed (W_1_) Falcon tube and centrifuged with a Universal 320 R centrifuge (Hettich, Tuttlingen, Germany), at 5550× *g* for 5 min. After discarding the supernatant, the sediment was dried at 80 °C to a constant weight in a universal laboratory oven (UN 75, Memmert, Schwabach, Germany). The dried biomass accumulation (g/L) of microalgae was assessed in triplicate using the following formula [123]:W = (W_2_ − W_1_) × 1000/V(1)
where W_1_ represents the empty Falcon tube weight (g), W_2_ represents the Falcon tube weight, including microalgae-dried biomass, and V represents the culture volume tested. The biomass was used for the determination of the soluble proteins.

#### 4.6.7. Pigment Concentrations after Extraction

For pigment measuring, we used the method described by Chai [125]. Briefly, 2 mL of each sample was centrifuged at 7400× *g* for 3 min. Over the remaining pellet, we added 2 mL of DMSO preheated to 60 °C and vortex for 10 min. Before reading the absorbance with a UV-Vis spectrophotometer (CLARIOstar, BMG Labtech, Ortenberg, Germany), the Eppendorf tubes were centrifuged using the same parameters mentioned above. We measured the absorbance at three different wavelengths, 480 nm, 649 nm, and 665 nm, necessary for the calculation of the content of chlorophyll *a*, chlorophyll *b*, total carotenoids, and total pigments in the samples, according to the following formulas:Chlorophyll *a* (ChlA) (mg/L) = 12.47 × (OD665) − 3.62 × (OD649)(2)
Chlorophyll *b* (ChlB) (mg/L) = 25.06 × (OD649) − 6.5 × (OD665)(3)
Total carotenoid (mg/L) = [1000 (OD480) − 1.29 (ChlA) − 53.78 (ChlB)]/220(4)
Total pigments (mg/L) = (1) + (2) + (3)(5)

#### 4.6.8. Protein Content

For protein extraction, we used freeze-dried biomass obtained after 14 days of cultivation. Basically, for solubilization, we left 5 mg microalgae powder suspended in 50 μL of 1 M NaOH for 10 min in a water bath at 80 °C. Then, we added 450 μL of distilled water, and the resulting solution was centrifuged for 30 min at 12,000× *g* (in a Universal 320 R centrifuge, Hettich, Tuttlingen, Germany), and the supernatant was moved to another tube. The above procedure was repeated twice, and the last extract was combined. The protein content was determined using two methods, Bradford and Biuret. For the Bradford method, we used bovine serum albumin (BSA) as standard in a concentration series of 0–10 µg/mL [95]. Briefly, over 200 µL of a sample, we added 50 µL of Bradford Reagent (Bradford Reagent, Sigma, Merck Group, Darmstadt, Germany), incubated for 5 min, and read the absorbance in a 96 well plate reader (CLARIOStar, BMG Labtech, Ortenberg, Germany) at a wavelength λ of 595 nm. For Biuret, the BSA standard curve was 0–10 mg/mL. Briefly, we added 200 µL of Biuret reagent to 40 µL of the sample; we incubated it in the dark at room temperature for 30 min, and then the absorbance (CLARIOStar, BMG Labtech, Ortenberg, Germany) was read at the wavelength λ = 555 nm.

#### 4.6.9. Chlorophyll Fluorescence

The chlorophyll fluorescence was measured using a handheld PAM fluorometer (PSI AquaPen AP 110/P, Photon Systems Instruments, Drásov, Czech Republic), according to manufacturer instructions, after microalgae adaptation to dark for 5 min. We calculated the maximum quantum yield of PSII photochemistry φP_0_ according to the following formula [126]:φP_0_ = F*v*/F*m*(6)
where F*v*—maximum variable fluorescence, F*m*-F*o*; F*m*—maximum chlorophyll *a* fluorescence (after actinic flash); F*o*—minimum chlorophyll *a* fluorescence (after dark adaptation).

### 4.7. Statistical Analysis

The data were subjected to one-way ANOVA (with significance *p* ≤ 0.05) data analysis, and the mean differences were compared using the Tukey HSD post hoc test. All studies were conducted in triplicate, and the values were expressed as mean ± standard error. The statistical analysis was performed using IBM SPSS statistics software (SPSS version 26.0)

## 5. Conclusions

Novel strigolactone mimics that contain aryloxy groups or a benzothiazole scaffold linked to the bioactive 3-methyl-5*H*-furan-2-one unit have been synthesized and characterized. The first two strigolactone mimics bearing aryloxy groups were obtained by the coupling reactions of the corresponding 3,5-disubstituted phenols with 5-bromo-3-methyl-5*H*-furan-2-one in the presence of a basic catalyst. Three new synthetic strigolactone mimics in which the nitrogen atom from the benzothiazole nucleus is directly linked to 3-methyl-5*H*-furan-2-one ring are the results of the *N*-akylating reactions of the keto-tautomeric forms of 2-hydroxybenzothiazole derivatives, namely 3*H*-benzothiazol-2-one, 6-bromo-3*H*-benzothazol-2-one and 6-nitro-3*H*-benzothazol-2-one, respectively, with 5-bromo-3-methyl-5*H*-furan-2-one.

The novel strigolactone mimics modulate the photosynthesis and biomass accumulation in *Ch. sorokiniana* NIVA-CHL 176 and could be considered microalgae biostimulants. Our experiments demonstrate that their effects include stimulation of biomass accumulation (as a result on an enhanced nutrient uptake and nutrient use), stimulation of the photoprotective photosynthetic pigments (chlorophyll *b*) synthesis, and enhanced quantum yield efficiency in photosystem II (as a result of increased tolerance to the stress resulted from the application of light for long photoperiod) and improved biomass quality, due to the increased content of soluble proteins.

Beside the practical importance, as a tool to improve exploitation of microalgae biotechnology potential, our results suggest that the perception of strigolactones by microalgae is similar to that of other organisms, being related to the methylated butenolide D-ring.

## Data Availability

X-ray diffraction data can be obtained free of charge via https://www.ccdc.cam.ac.uk/structures/, accessed on 10 September 2023 (or from the Cambridge Crystallographic Data Centre, 12 Union Road, Cambridge CB2 1EZ, UK; Fax: (+44) 1223-336-033; or deposit@ccdc.ca.ac.uk).

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
