# Peer review of "Novel Strigolactone Mimics That Modulate Photosynthesis and Biomass Accumulation in Chlorella sorokiniana"

_molecules, 2023, doi:10.3390/molecules28207059_

Round 1

Reviewer 1 Report

This work aims to synthesize and identify strigolactone mimics that promote photosynthesis and biomass accumulation in microalgae with biotechnological potential. The authors found some valuable conclusions, such as the SL-F3 mimic, 3-(4-methyl-5-oxo-2,5-dihydrofuran-2-yl)-3H-benzothiazol-2-one (7) proved the most efficient. This topic is meaningful. Thus, I recommend this manuscript be considered for publication after some revisions.

1. In the discussion of data (Fig. 4- Fig. 9), two concentrations of 10-7 M and 10-9 M were used. What is the rationale for choosing these two concentration gradients? Please explain in the manuscript.

2. When exploring changes in optical density of microalgae cultures over time, why were only 2 to 14 days chosen? Lines 217-218, “At the end of the culturing period, after 14 days, the cell growth upon some treatments was significantly higher”, Does this mean that 14 days has not reached the end of the culturing period?

3. The legend colors of Figure 4 are similar (such as dark gray day9 and light gray day5). It is recommended to make the following changes to enhance the color contrast.

4. There are too many acronyms in the manuscript, and it is suggested that a table be added listing the full names of these acronyms so that the reader can read them better.

5. Line 348. “Our target was to prepare novel strigolactone mimics easily accessible in significant amounts from simple and available starting materials”. The use of first person (we, our, us) should be avoided.

6. Line 452-464. It seems that the concentration of each of the BG11 ingredients is more important than where it is purchased. It is recommended that this section be amended accordingly.

Reviewer 2 Report

This paper contains a detailed account of preparation and testing of a number of small molecules for evaluation as biomass accumulation modulators. The natural products which act as starting points, the stringolactones, are very complex in structure. The authors provide a very detailed and comprehensive introduction to the field in the paper. The model compounds in the paper are much simpler and were prepared by the authors through a simple and logical one-step process. These were characterised to a good level of detail, and the full structure of one example was confirmed by X-ray crystallography. Although much simpler in structure, the compounds exhibit some promising activity, with microalgae biostimulant activity detected for several compounds even at low concentrations of 10 -9 M. I am not expert in this yield, therefore an expert might better understand the context, however the results appear to be very convincing and will be of significant interest to researchers in this area.

Although the synthetic work is based on previously published methodology, it is usefully extended here and is not a major component of the paper, although it is described in a good level of detail. The majority of the work is focussed on the evaluation of the biological properties of the molecules, and this is quite extensive and valuable. The paper contains work of good quality overall. The experimental section and supporting information are of sufficient clarity to allow the work to be repeated, and small molecules are appropriately characterised, including by CHN analysis. The references are cited appropriately, with DOIs included in most cases.

I recommend publication of the paper, however there are a few minor issues to attend to;

In the SI there are a number of NMR spectra but they are rather weak and give poor resolution relative to the baseline. Given that the X-ray of 7 was obtained, it is surprising that a cleaner NMR sample was not obtained. Whilst not essential the authors can obtain improved versions then they will increase the quality of the paper.

In the NMR spectra, the integral lines should be clear of the peak picking ones, for clarity.

Scheme 1 and other synthetic schemes add yields to the schemes.

One thing that is confusing in the paper is that the mimic which are prepared – numbered 3, 4 7, 9 and 10 – are then renumbered as ‘SL-F1-5’ which are then applied to the biological effects graphs. It might be better if the SL-F1-5’ names were used from the outset so that there is no ambiguity in the labelling.

The paper is generally well-written however a careful check by an expert proof reader is recommended, to identify any minor e,g. typographical, errors.

Reviewer 3 Report

About Figure 1, Scheme 1

Why the researchers didn’t mention the source of this figure and get permission from the source?

About Figure 9

The differences between different treatments are few despite the statistical analysis ... for example here 0.75 mg.

About Materials and Methods

It is possible to take sub-sample from Chlorella sorokiniana NIVA- CHL-176 strain.

About B11 – no need to give more details. Enough to mention the reference.

In line 595

4.6.5. Growth parameters – Need References
